# A Self-Immolative Linker for the pH-Responsive Release of Amides

**DOI:** 10.3390/molecules28062445

**Published:** 2023-03-07

**Authors:** Agnese Petrini, Giovanni Ievoli, Francesca Migliorini, Maurizio Taddei, Sofia Siciliano

**Affiliations:** Dipartimento di Biotecnologie, Chimica e Farmacia Università di Siena, Via A. Moro 2, 53100 Siena, Italy

**Keywords:** drug delivery, bioconjugation, self-immolative spacer, stimuli sensitive systems, amides

## Abstract

The administration of therapeutics using bioconjugation has been mainly limited to drugs containing amine, alcohol, or thiol functional groups. Here, we report a general procedure for the preparation of benzylic N-acyl carbamates suitable for masking the amide group in important drugs such as Linezolid, Enzalutamide, or Tasimelteon in good to acceptable yields. These N-acyl carbamates appear to be stable in plasma, while a qualitative analysis of further drug uncage demonstrates that, at pH values of 5.5, a classical 1,6-benzyl elimination mechanism takes place, releasing more than 80% of the drug in 24 h.

## 1. Introduction

Bioconjugation has recently attracted remarkable interest thanks to its therapeutic potential in chemotherapy [1,2,3]. As selectivity is one of the major problems in the use of cytotoxic drugs, bioconjugation provides a powerful tool for smart drug delivery [4,5,6,7]. The active pharmaceutical ingredient can be covalently bound to selective carriers such as small molecule ligands, peptides, proteins, or antibodies. Once the bioconjugate has reached the target, the drug can be triggered by an external stimulus. The linker is a key component of the bioconjugate, as it not only separates the carrier from the drug, but also controls stability in the bloodstream and release in the target tissue [1,6,8,9]. Two main families of linkers are used in bioconjugation. The so-called “non-cleavable linker” [10] acts through the lysosomal proteolytic degradation of the conjugate, while the “cleavable linker” [11] relies on endogenous stimuli such as enzyme activity, pH, high ROS, or glutathione, which activate a self-immolative degradation that ends with the release of the intact drug [8]. However, this self-immolative process relies on molecules with high nucleofugacity, which limits the release to agents containing amines, alcohols, phenols, or thiols (Figure 1, reaction a) [8,12]. Amides are widely distributed molecules in nature, and several approved drugs contain this functional group [13]. Despite their importance, there are very few bioreversible prodrugs or self-immolative linkers suitable for the targeted release of amides [14,15,16,17,18]. This is because of the low nucleophilicity of the amide nitrogen and the low acidity of the amide hydrogen, which prevent nucleofugacity.

Here, we decided to investigate the use of N-acyl carbamates to develop a new self-immolative linker for amides (Figure 1, reaction b). The new prodrug is an N-acyl carbamate derived from *p*-hydroxybenzyl alcohol and protected on the phenol side with an acid-labile group. As pH variations between healthy and cancerous or inflamed tissues have been used to produce pH-sensitive materials [19,20,21,22,23,24], we expected the drug to be released by acid-mediated degradation. Acyl carbamates have been described as potential prodrugs for amides since the 1990s [15]. Although they are relatively stable at pH between 4 and 6 and appear to be stable in plasma for several hours [25], no significant applications have been described to date [26].

## 2. Results

We developed our linker by adding acid-labile protecting groups to the phenolic side of the p-hydroxybenzyl alcohol, while the benzyl alcohol served to install the amide-containing drug over the acyl carbamate. In addition, a terminal alkyne was added to the aromatic ring to allow coupling with the carrier (Figure 2). Linezolid (7 in Figure 2) was used as a model drug to optimise linker design and synthesis. It is an antibiotic used to treat infections caused by Gram-positive resistant bacteria and contains only a secondary amide as a functional group suitable for conjugation [27]. 3,4-Dihydroxybenzaldehyde **1** was selectively protected at position 4 as methoxymethyl ether (MOM, 2a in Figure 2) or methoxyethoxymethyl ether (MEM, 2b in Figure 2) in acceptable yields. The other phenol group was alkylated with propargyl bromide to introduce a counterpart suitable for further conjugation by click chemistry (**3a**,**b** in Figure 2). Reduction of the aldehyde gave the benzyl alcohols **4a**,**b**, the starting materials for N-acyl carbamate synthesis. N-Acyl carbamates have been prepared mainly by addition of an alcohol to an acyl isocyanate obtained by treatment of a primary amide with oxalyl chloride [17,18]. Alternatively, displacement of an activated carbonate by a primary amide has been described [15]. However, these methods are not suitable for the preparation of an acyl carbamate prodrug of Linezolid such as **8a** or **8b** in Figure 2. We investigated the possibility of activating compounds **4a**,**b** with *p*-NO_2_-phenylchloroformic acid ester to obtain carbonates **5a**,**b**. These compounds were relatively stable molecules and could be purified by flash chromatography. Compound **5a** was used to study the direct oxycarbonylation of Linezolid **7** or the indirect acylation of amine 6 followed by acetylation to give the Linezolid adduct **8a**. Treatment of **7** with **5a** in the presence of different bases and different solvents was unsuccessful and gave only small amounts of 8a (entries 1–4 in Table 1). Better results were obtained by carbamoylation of amine **6**, which was carried out in good yields (entry 5 in Table 1). Unfortunately, acetylation of the resulting carbamate with Ac_2_O, DMAP, and Et3N gave compound **8a** in a low yield (entry 5 in Table 1). Further attempts to increase the yield by changing the base were unsuccessful.

The acyl carbamate **8a** was finally obtained in a good yield (88%) by deprotonation of the amide NH with KHMDS in THF at −78 °C, followed by reaction with carbonate **5a** in THF at a controlled temperature (entry 6 in Table 1). The same procedure was applied to **5b**, giving **8b** in a 59% isolated yield (Figure 2). In the latter case, the low yield was due to the fact that the carbonate **8b** tends to degrade after 5/6 h under alkaline conditions. Consequently, the unreacted amide **7** was recovered at the end of the reaction. Acylation of an amide with benzyl carbonate **5a** in the presence of KHMDS was also used with Enzalutamide **9** (Figure 2), a non-steroidal antiandrogen approved for the treatment of prostate cancer [28]. 

Again, the amide is the only functional group in the molecule available for bioconjugation and further release of the drug under an endogenous stimulus. Following the general procedure developed for **8**, Enzalutamide pro-drug **10** was obtained in 64% isolated yield (Figure 2). Products **8a**,**b** and **10** were tested for stability in solution and were stable for 48 h at a pH of 7.4 (PBS solution). Unfortunately, at a pH of 5.5, only a small amount of amide **9** was released after 48 h at 37 °C. A small improvement was achieved at a pH of 4.5, where the peak intensities for **8a**,**b** and **10** decreased, but only the presence of free Enzalutamide **9** (40%) could be clearly detected. These data show that it is possible to benzoxylate the potassium salt of a (secondary) amide with a *p*-nitrophenyl carbonate such as **5a**,**b** to give an acyl carbamate stable in PBS solution. However, the use of MEM or MOM to protect the phenolic groups, already used in pro-drug applications [29], prevents the use of these conjugates for self-immolative systems triggered by physiological small pH fluctuations [30].

Therefore, we decided to form the activated carbonate from benzyl alcohol **11** (Figure 3). This linker, which is based on orthoester chemistry, was derived from gallic acid and is known to undergo the uncage process at a pH of 5.5 [31]. Carbonylation of **11** was carried out as previously described for compound **4**, and the activated carbonate **12** was used for the acylation of Linezolid **7**, Enzalutamide **9**, and Tasimelteon **13**, a drug developed for sleep disorders [32]. *p*-Toluenesulfonamide **14** was also used as a model compound for sulfonamide-containing drugs. Amides **7**, **9**, and **13** were chosen as representative of approved drugs with different therapeutic applications where the NH amide is the only nucleophile for linkage. To the best of our knowledge, there are no examples in the literature of prodrugs or bioconjugates that release these drugs.

Following the synthetic approach described in Figure 3, N-acyl carbamates **15**–**18** were obtained in good to acceptable yields.

The low yield of compound **17** (44%) was due to the degradation of Tasimelteon during the reaction, while the low yield of compound **18** (21%) was due to the sluggish reactivity characterised by the recovery of unreacted starting material and some non-separable mixed fractions after column chromatography.

Qualitative analysis showed that products **15**–**18** appear stable in physiological fluids and proved stable in PBS and human plasma, as indicated in Table 2. All products were stable in water, while we observed some decrease in product peaks after 24 h in PBS and in plasma (Table 2 columns 3 and 4). However, the presence of the free drugs was never detected.

A qualitative analysis of the hydrolysis of compounds **15**–**18** was also performed and it was found that the release of the intact drug amides **7**, **9**, **13**, and **14** occurred at pH 5.5 (Figure 1 and Appendix A). In the case of the more acidic *p*-toluenesulfonamide derivative, release was faster, but after 48 h, no more than 80% had been released.

## 3. Conclusions

In summary, we have shown that the acid-sensitive benzyl alcohols **4a**,**b** and **11** can be used to mask drugs containing an amide bond by forming the corresponding N-acyl carbamate, although they are restricted to low hindered amides. Orthogonal reactivity to this platform is possible by click chemistry to allow easy conjugation with macromolecular carriers. Our system appears stable in plasma, allowing potential application in targeted drug delivery. The intact drug amide is released after triggering a 1–6 elimination cascade in an acidic biological environment at a pH typical of inflamed tissue, or by lysosomal digestion in the case of application of the linker in ADC. Compounds **15**–**18** are the first successful examples of how a drug can be linked to a carrier with an amide to create a bioconjugate or pro-drug or nanomedical device that releases the drug once it reaches the target. Further improvements to extend this linker to more complex amides using a different synthetic approach and using other non-canonical chemotherapeutic agents triggered by enzymatic release processes are in preparation and will be announced shortly.

## 4. Materials and Methods

General experimental procedures, materials, and instruments are reported in Appendix A.

**3-Hydroxy-4-(methoxymethoxy)benzaldehyde 2a.** The product was prepared according to the literature [33]. Under an N_2_ atmosphere, 3,4-dihydroxybenzaldehyde **1** (700 mg, 5.07 mmol) was suspended in CH_3_CN dry (25 mL) and K_2_CO_3_ (2.10 g, 15.20 mmol) was added. The mixture was stirred for 30 min at RT and chloromethyl methyl ether (0.30 mL, 2.1 M in toluene, 6.30 mmol) was added. After stirring for 16 h at RT, NaOH 10% (25 mL) and EtOAc (3 × 25 mL) were added and the aqueous phase extracted, neutralized with HCl 1 N until pH 9, and extracted again with EtOAc (2 × 25 mL). The organic phases were collected and washed with Na_2_CO_3_ (1 × 25 mL) and brine (1 × 25 mL), dried over anhydrous Na_2_SO_4_, filtered, and evaporated under reduced pressure. The final crude product (593 mg, 64% yield) was obtained as a yellow oil. ^1^H NMR (400 MHz, CDCl_3_): δ 9.75 (s, 1H), 7.38 (s, 1H), 7.31 (d, *J* = 8.2 Hz, 1H), 7.13 (d, *J* = 8.3 Hz, 1H), 5.75–6.5 (bs, 1H), 5.22 (s, 2H), 3.43 (s, 3H). ^13^C NMR (101 MHz, CDCl_3_) δ 191.5, 149.9, 146.7, 131.3, 124.3, 114.9, 114.3, 95.1, 56.6. ESI-MS: *m*/*z* 205 [M + Na]^+^.

**4-(Methoxymethoxy)-3-(prop-2-yn-1-yloxy)benzaldehyde 3a** Compound **2a** (500 mg, 2.75 mmol) was solubilized in dry acetone (15 mL) under an N_2_ atmosphere and mixed with K_2_CO_3_ (1.13 g, 8.24 mmol) and KI (457 mg, 2.75 mmol). After 10 min, propargyl bromide (981 mg of a 30% solution in toluene, 8.24 mmol) was added and the reaction was refluxed for 16 h. Acetone was evaporated and the crude was solubilized in CHCl_3_ (50 mL) and washed with H_2_O (2 × 25 mL) and brine (25 mL), and the organic phase was dried over anhydrous Na_2_SO_4_, filtered, and evaporated under reduced pressure. The product was purified by chromatography on silica gel with MPLC (medium pressure liquid chromatography), 0–50% gradient of EtOAc in petroleum ether to obtain the product **3a** as a pale-yellow oil (417 mg, 68% yield). ^1^H NMR (400 MHz, CDCl_3_): δ 9.73 (s, 1H), 7.44 (d, *J* = 1.1 Hz, 1H), 7.35 (dd, *J* = 8.3, 1.1 Hz, 1H), 7.15 (d, *J* = 8.3 Hz, 1H), 5.18 (s, 2H), 4.70 (d, *J* = 1.9 Hz, 2H), 3.38 (s, 3H), 2.53–2.44 (m, 1H). ^13^C NMR (101 MHz, CDCl3) δ 190.8, 152.5, 147.8, 130.9, 126.8, 115.2, 112.7, 95.04, 77.7, 76.4, 56.6, 56.6. ESI-MS: *m*/*z* 243 [M + Na]^+^. HRMS calcd for C_12_H_12_NaO_4_ [M + Na]^+^ 243.0633; found 243.0631.

**(4-(Methoxymethoxy)-3-(prop-2-yn-1-yloxy)phenyl)methanol 4a.** NaBH_4_ (373 mg, 9.86 mmol) was suspended in dry MeOH (11 mL) under an N_2_ atmosphere and cooled at 0 °C. Then, a solution of compound **3a** (542 mg, 2.46 mmol) in dry CH_2_Cl_2_ (30 mL) was slowly added and the reaction mixture was stirred for 2 h at 0 °C. Water was added until a white suspension appeared and the mixture was extracted with CH_2_Cl_2_ (3 × 25 mL). The organic phases were washed with H_2_O (25 mL) and brine (25 mL), dried over anhydrous Na_2_SO_4_, filtered, and evaporated under reduced pressure. The product was purified by silica gel MPLC, 0–60% gradient of EtOAc in petroleum ether to obtain the product (466 mg, 85% yield). ^1^H NMR (400 MHz, CDCl_3_): δ 7.10 (d, *J* = 8.2 Hz, 1H), 7.06 (d, *J* = 1.7 Hz, 1H), 6.91 (dd, *J* = 8.2, 1.7 Hz, 1H), 5.18 (s, 2H), 4.75 (s, 2H), 4.60 (d, *J* = 2.4 Hz, 2H), 3.48 (s, 3H), 2.48 (t, *J* = 2.4 Hz, 1H), 1.62 (bs, 1H). ^13^C NMR (101 MHz, CDCl_3_): δ 147.4, 146.1, 135.0, 120.3, 116.6, 113.3, 95.2, 78.1, 75.4, 64.5, 56.4, 55.8. ESI-MS: *m*/*z* 245 [M + Na]^+^. Anal. calcd. for C_12_H_14_O_4_ C 64.85, H 6.35, O 28.80; found C 64.81, H 6.33.

**4-(Methoxymethoxy)-3-(prop-2-yn-1-yloxy)benzyl (4-nitro-phenyl) carbonate 5a.** Compound **4a** (78 mg, 0.35 mmol) was solubilized in dry THF (8 mL) and cooled at 0 °C. DMAP (4 mg, 0.04 mmol), 4-nitrophenyl chloroformate (142 mg, 0.70 mmol), and pyridine (57 µL, 0.7 mmol) were added and the reaction was stirred at RT for 16 h. THF was evaporated and the crude was dissolved in EtOAc (20 mL) and washed with H_2_O (10 mL), brine (10 mL), dried over anhydrous Na_2_SO_4_, filtered, and evaporated under reduced pressure. The product was purified by silica gel MPLC, 0–30% gradient of EtOAc in petroleum ether to obtain the product (112 mg, 82% yield). ^1^H NMR (400 MHz, CDCl_3_) δ 8.24 (d, *J* = 9.2 Hz, 2H), 7.35 (d, *J* = 9.2 Hz, 2H), 7.15 (d, *J* = 8.3 Hz, 1H), 7.11 (d, *J* = 1.9 Hz, 1H), 7.02 (dd, *J* = 8.3, 1.9 Hz, 1H), 5.21 (s, 4H), 4.76 (d, *J* = 2.4 Hz, 2H), 3.49 (s, 2H), 2.48 (t, *J* = 2.4 Hz, 1H). HRMS calcd for C_19_H_17_NNaO_8_ [M + Na]^+^ 410.0852, found 410.0850.

**3-Hydroxy-4-((2-methoxyethoxy)methoxy)benzaldehyde 2b.** Obtained as previously described for **2a.** Obtained 711 mg of a pale yellow oil, 83% yield. ^1^H NMR (400 MHz, CDCl_3_): δ 9.82 (s, 1H), 7.42 (d, *J* = 1.9 Hz, 1H), 7.35 (dd, *J* = 8.3, 2.0 Hz, 1H), 7.17 (d, *J* = 8.3 Hz, 1H), 6.5–6.0 (bs, 1H), 5.34 (s, 2H), 3.84 (dd, *J* = 5.2, 3.7 Hz, 2H), 3.55 (dd, *J* = 5.4, 3.5 Hz, 2H), 3.36 (s, 3H). ^13^C NMR (101 MHz, CDCl_3_) δ 191.2, 149.9, 147.1, 131.7, 124.0, 115.2, 115.0, 94.8, 71.5, 68.8, 59.0. ESI-MS: *m*/*z* 249 [M + Na]^+^ HRMS Calcd for C_11_H_14_NaO_5_ [M + Na]^+^ 249.0739, found 249.0737.

**4-((2-Methoxyethoxy)methoxy)-3-(prop-2-yn-1-yloxy)benz-aldehyde 3b.** Obtained as previously described for **3a.** MPLC 0–60% gradient of EtOAc in petroleum ether gave the product **3b** as a yellow oil (335 mg, 80% yield). ^1^H NMR (400 MHz, CDCl_3_): δ 9.70 (s, 1H), 7.40 (s, 1H), 7.33 (d, *J* = 8.2 Hz, 1H), 7.17 (d, *J* = 8.3 Hz, 1H), 5.24 (s, 2H), 4.67 (d, *J* = 2.1 Hz, 2H), 3.72 (d, *J* = 4.3 Hz, 2H), 3.40 (d, *J* = 4.5 Hz, 2H), 3.19 (s, 3H), 2.48 (s, 1H).^13^C NMR (101 MHz, CDCl_3_): δ 190.7, 152.4, 147.8, 130.8, 126.8, 115.5, 112.5, 93.9, 77.8, 76.5, 71.4, 68.2, 58.9, 56.6. ESI-MS: *m*/*z* 287 [M + Na]^+^. HRMS Calcd for C_14_H_16_NaO_5_ [M + Na]^+^ 287.0895, found 287.0893.

**(4-((2-Methoxyethoxy)methoxy)-3-(prop-2-yn-1-yloxy)-phenyl)methanol 4b.** Obtained as previously described for **4a.** MPLC 0–70% gradient of EtOAc in petroleum ether gave the product **4b** as a colourless oil (162 mg, 78% yield). ^1^H NMR (400 MHz, CDCl_3_): δ 7.07 (d, *J* = 8.2 Hz, 1H), 6.97 (s, 1H), 6.83 (d, *J* = 8.2 Hz, 1H), 5.19 (s, 2H), 4.66 (s, 2H), 4.50 (s, 2H), 3.83–3.73 (m, 2H), 3.49–3.43 (m, 2H), 3.28 (s, 3H), 2.56 (s, 1H), 2.45 (s, 1H). ^13^C NMR (101 MHz, CDCl_3_): δ 147.3, 146.1, 135.1, 120.3, 116.7, 113.2, 94.2, 78.1, 75.4, 71.1, 67.4, 64.3, 58.5, 56.3. ESI-MS: *m*/*z* 289 [M + Na]^+^. HRMS calcd for C_14_H_18_NaO_5_ [M + Na]^+^ 289.1052, found 289.1054.

**4-((2-Methoxyethoxy)methoxy)-3-(prop-2-yn-1-yloxy)benzyl (4-nitrophenyl) carbonate 5b.** Obtained as previously described for **4a**. MPLC 0–50% gradient of EtOAc in petroleum ether gave the product **5b** as a colourless oil (198 mg, 82% yield). ^1^H NMR (400 MHz, CDCl_3_): δ 8.20 (d, *J* = 9.0 Hz, 2H), 7.32 (d, *J* = 9.0 Hz, 2H), 7.17 (d, *J* = 8.3 Hz, 1H), 7.08 (s, 1H), 6.99 (d, *J* = 8.3 Hz, 1H), 5.27 (s, 2H), 5.18 (s, 2H), 4.73 (d, *J* = 2.2 Hz, 2H), 3.85–3.78 (m, 2H), 3.54–3.48 (m, 2H), 3.32 (d, *J* = 3.7 Hz, 3H), 2.48 (t, *J* = 2.0 Hz, 1H). ESI-MS: *m*/*z* 432 [M + H]^+^. HRMS calcd for C_21_H_22_NO_9_ [M + H]^+^. 432.1295, found 431.1297.

**General procedure for the synthesis of carbamates. {[3-(3-Fluoro-4-morpholinophenyl)-2-oxo-1,3-oxazolidin-5-yl]methyl}acetylamino-{4-[(2-methoxyethoxy)methoxy]-3-(2-propynyloxy)-phenyl}methylformylate 8b.** Linezolid (50 mg, 0.15 mmol) was solubilized in dry THF (0.8 mL) under an N_2_ atmosphere in a Schlenk tube and the mixture was cooled at −78 °C. Then, KHMDS (0.15 mL of a 0.5 M solution in toluene, 0.13 mmol) was added and the reaction was stirred at −78 °C for 1 h. A solution of carbonate **5b** (52 mg, 0.13 mmol) in 0.8 mL of dry THF was slowly added, the temperature was raised from −78 °C to −5 °C, and the mixture was stirred at this temperature for 4 h. Then, EtOAc (15 mL) was added; the mixture was warmed to room temperature; and the organic phase washed with NH_4_Cl (10 mL), H_2_O (10 mL), and brine (10 mL) The organic layer was dried over anhydrous Na_2_SO_4_, filtered, and evaporated under reduced pressure. Compound **8b** was purified through silica gel MPLC, 0–70% gradient of EtOAc in petroleum ether (42 mg, 59% yield). ^1^H NMR (400 MHz, CDCl_3_): δ 7.35 (d, *J* = 14.3 Hz, 1H), 7.15 (dd, *J* = 8.2, 2.1 Hz, 1H), 7.07 (s, 1H), 7.02 (d, *J* = 8.8 Hz, 1H), 6.96 (d, *J* = 8.2 Hz, 1H), 6.89 (dd, *J* = 12.7, 5.4 Hz, 1H), 5.30–5.09 (m, 4H), 4.73 (s, 3H), 4.22–3.94 (m, 4H), 3.83 (d, *J* = 3.3 Hz, 6H), 3.67–3.58 (m, 1H), 3.54–3.50 (m, 2H), 3.33 (d, *J* = 2.2 Hz, 2H), 3.05–2.95 (m, 4H), 2.52 (d, *J* = 2.1 Hz, 3H), 2.46 (d, *J* = 2.3 Hz, 1H). ^13^C NMR (101 MHz, CDCl_3_): δ 172.9, 156.3, 153.9, 153.6, 147.3, 136.1, 132.6, 128.2, 122.6, 118.4, 116.5, 114.9, 113.5, 107.2, 106.9, 94.1, 78.0, 75.4, 71.1, 70.1, 68.8, 67.5, 66.5, 58.5, 56.4, 50.6, 48.0, 46.3, 26.2. ESI-MS: *m*/*z* 630 [M + H]^+^, 652 [M + Na]^+^. Anal calcd for C_31_H_36_FN_3_O_10_: C 59.13, F 3.02, H 5.76, N 6.67, O, 25.41; found C 59.10, H 5.74, N 6.69.

**4-(Methoxymethoxy)-3-(prop-2-yn-1-yloxy)benzyl (R)-acetyl((3-(3-fluoro-4-morpholino-phenyl)-2-oxooxazolidin-5-yl)methyl)carbamate 8a.** Compound **8a** was prepared according to general procedure. The compound was purified through silica gel MPLC, 0–70% gradient of EtOAc in petroleum ether (51 mg, 88% yield). ^1^H NMR (400 MHz, CDCl_3_): δ 7.34 (dd, *J* = 14.3, 2.5 Hz, 1H), 7.09 (dd, *J* = 10.9, 5.0 Hz, 2H), 6.99 (ddd, *J* = 10.1, 8.5, 2.1 Hz, 2H), 6.88 (t, *J* = 9.1 Hz, 1H), 5.28–5.06 (m, 4H), 4.71–4.80 (m, 3H), 4.13 (dd, *J* = 14.2, 7.6 Hz, 1H), 4.08–3.92 (m, 2H), 3.88–3.79 (m, 4H), 3.61 (dd, *J* = 9.1, 5.7 Hz, 1H), 3.46 (d, *J* = 6.1 Hz, 3H), 3.12–2.96 (m, 4H), 2.51 (s, 3H), 2.46 (s, 1H). ^13^C NMR (101 MHz, CDCl_3_): δ 172.8, 156.2, 153.8, 153.5, 153.4, 147.3, 136.0, 132.7, 132.6, 128.1, 122.5, 118.4, 116.48, 115.0, 113.5, 107.1, 106.9, 95.0, 78.0, 75.5, 70.1, 67.5, 66.4, 56.4, 55.8, 50.5, 47.9, 46.2, 26.1. ESI-MS: *m*/*z* 586 [M + H]^+^; 608 [M + Na]^+^; 624 [M + K]^+.^ Anal calcd for C_29_H_32_FN_3_O_9_ C 59.48; F 3.24; H 5.51; N 7.18; O 24.59; found C 59.39, H 5.49, N 7.16.

**4-(Methoxymethoxy)-3-(prop-2-yn-1-yloxy)benzyl (4-(3-(4-cyano-3-(trifluoromethyl)-phenyl)-5,5-dimethyl-4-oxo-2-thioxoimidazolidin-1-yl)-2-fluorobenzoyl)(methyl)-carbamate 10.** Compound **10** was prepared according to general procedure for synthesis of carbamates. Time reaction: 4 h. The compound was purified through silica gel MPLC, 0–50% gradient of EtOAc in petroleum ether (46 mg, 64% yield). ^1^H NMR (400 MHz, CDCl_3_): δ 7.96 (s, 1H), 7.93 (d, *J* = 5.5 Hz, 1H), 7.80 (dd, *J* = 8.3, 1.8 Hz, 1H), 7.59 (t, *J* = 7.8 Hz, 1H), 7.14 (dd, *J* = 8.2, 1.7 Hz, 1H), 7.06 (d, *J* = 8.3 Hz, 1H), 6.98 (dd, *J* = 10.0, 1.7 Hz, 1H), 6.95 (d, *J* = 1.9 Hz, 1H), 6.81 (dd, *J* = 8.3, 1.9 Hz, 1H), 5.17 (d, *J* = 2.3 Hz, 2H), 5.02 (s, 2H), 4.71 (d, *J* = 2.4 Hz, 2H), 3.47 (s, 3H), 3.37 (s, 3H), 2.47 (t, *J* = 2.3 Hz, 1H), 1.56 (s, 6H) ^13^C NMR (101 MHz, CDCl_3_): δ 179.7, 174.5, 166.8, 159.9, 157.4, 153.9, 147.7, 147.5, 138.0, 137.9, 136.9, 135.3, 133.8, 133.5, 132.1, 130.2, 128.6, 127.4, 127.3, 127.1, 127.0, 125.7, 123.2, 122.9, 120.5, 117.3, 117.0, 116.7, 115.5, 114.7, 110.4, 95.4, 75.9, 68.7, 66.6, 57.0, 56.3, 32.1, 29.7, 23.8. ESI-MS: *m*/*z* 713 [M + H]^+^; 735 [M + Na]^+^. Anal calcd for C_34_H_28_F_4_N_4_O_7_S: C 57.30, F 10.66, H 3.96, N 7.86, O 15.71, S 4.50; found C 57.27, H, 3.95, N 7.88.

**2-Ethoxy-6-[(*p*-nitrophenoxycarbonyloxy)methyl]-4-(2-propynyloxy)-2H-1,3-benzodioxole 12.** Obtained as previously described for **4a**. MPLC, 0–50% gradient of EtOAc in petroleum ether gave **12** as a colourless oil (101 mg, 70% yield). ^1^H NMR (400 MHz, CDCl_3_) δ 8.24 (d, *J* = 8.9 Hz, 2H), 7.35 (d, *J* = 8.9 Hz, 2H), 6.89 (s, 1H), 6.73 (s, 1H), 6.68 (s, 1H), 5.16 (s, 2H), 4.80 (s, 2H), 3.74 (q, *J* = 7.0 Hz, 2H), 2.51 (s, 1H), 1.24 (t, *J* = 7.0 Hz, 3H). ESI-MS: *m*/*z* 438 [M + Na]^+^. HRMS calcd for C_20_H_17_NNaO_9_ 438.0801; found 438.0802.

**(2-Ethoxy-7-(prop-2-yn-1-yloxy)benzo[d][1,3]dioxol-5-yl)methyl acetyl(((R)-3-(3-fluoro-4-morpholinophenyl)-2-oxooxazolidin-5-yl)methyl)carbamate 15.** Compound **15** was prepared according to the general procedure. Time reaction: 1.5 h. The compound was purified by silica gel MPLC, 0–70% gradient of EtOAc in petroleum ether (44 mg, 72% yield). ^1^H NMR (400 MHz, CDCl_3_): δ 7.40–7.30 (m, 1H), 7.01 (t, *J* = 7.5 Hz, 1H), 6.91–6.83 (m, 2H), 6.69 (s, 1H), 6.64 (d, *J* = 4.0 Hz, 1H), 5.12 (m, 2H), 4.78 (s, 2H), 4.12 (m, 1H), 4.00 (m, 2H), 3.89–3.79 (m, 4H), 3.71 (dt, *J* = 13.8, 7.0 Hz, 2H), 3.66–3.57 (m, 1H), 3.04–2.97 (m, 4H), 2.51 (s, 3H), 2.48 (d, *J* = 1.1 Hz, 1H), 1.59 (s, 1H), 1.27–1.18 (m, 3H). ^13^C NMR (101 MHz, CDCl_3_): δ 172.9, 156.3, 153.9, 153.6, 147.3, 140.2, 136.0, 134.5, 132.8, 128.2, 119.4, 118.5, 113.5, 110.7, 107.3, 103.2, 77.8, 75.6, 70.0, 68.9, 66.5, 59.2, 57.0, 50.6, 48.0, 46.3, 26.2, 14.3. ESI-MS: *m*/*z* 614 [M + H]^+^; 636 [M + Na]^+^. Anal calcd for C_30_H_32_FN_3_O_10_: C 58.72, F 3.10, H 5.26, N 6.85, O 26.08; found: C 58.70, H 5.25, N 6.87.

**(2-Ethoxy-7-(prop-2-yn-1-yloxy)benzo[d][1,3]dioxol-5-yl)methyl (4-(3-(4-cyano-3-(trifluoromethyl)phenyl)-5,5-dimethyl-4-oxo-2-thioxoimidazolidin-1-yl)-2-fluorobenzoyl)(methyl)carbamate 16.** Compound **16** was prepared according to the general procedure. Time reaction: 3 h. The compound was purified by silica gel MPLC 0–60% gradient of EtOAc in petroleum ether (44 mg, 61% yield). ^1^H NMR (400 MHz, CDCl_3_): δ 7.96 (s, 1H), 7.93 (d, *J* = 4.8 Hz, 1H), 7.80 (d, *J* = 8.3 Hz, 1H), 7.60 (t, *J* = 7.7 Hz, 1H), 7.15 (d, *J* = 8.1 Hz, 1H), 7.00 (d, *J* = 10.0 Hz, 1H), 6.83 (s, 1H), 6.55 (s, 1H), 6.48 (s, 1H), 4.97 (s, 2H), 4.76 (d, *J* = 1.1 Hz, 2H), 3.71 (q, *J* = 7.0 Hz, 2H), 3.37 (s, 3H), 2.48 (d, *J* = 2.2 Hz, 1H), 1.56 (s, 6H), 1.22 (t, *J* = 7.1 Hz, 3H). ^13^C NMR (101 MHz, CDCl_3_): 179.7, 174.5, 166.8, 159.9, 157.4, 153.8, 147.6, 140.4, 138.1, 138.0, 136.9, 135.3, 134.7, 132.1, 130.2, 128.7, 127.3, 127.2, 127.1, 127.0, 125.7, 119.7, 117.3, 117.1, 114.7, 111.1, 110.3, 103.4, 76.1, 68.8, 66.6, 59.8, 57.5, 32.1, 23.8, 14.8. ESI-MS: *m*/*z* 741 [M + H]^+^, 764 [M + Na]^+^. Anal calcd for C_35_H_28_F_4_N_4_O_8_S: C 56.75, F 10.26, H 3.81, N 7.56, O 17.28, S 4.33; found C 56.71, H 3.78, N 7.59.

**(2-Ethoxy-7-(prop-2-yn-1-yloxy)benzo[d][1,3]dioxol-5-yl)me-thyl(((1R,2R)-2-(2,3-dihydrobenzofuran-4-yl)cyclopropyl)-methyl)(propionyl)carbamate 17.** Compound **17** was prepared according to the general procedure. Time reaction: 3.5 h. The compound was purified by silica gel MPLC, 0–30% gradient of EtOAc in petroleum ether (23 mg, 44% yield). ^1^H NMR (400 MHz, CDCl_3_): δ 6.95 (t, *J* = 7.8 Hz, 1H), 6.86 (s, 1H), 6.66 (s, 1H), 6.61 (s, 1H), 6.55 (d, *J* = 7.9 Hz, 1H), 6.23 (dd, *J* = 7.7, 2.7 Hz, 1H), 5.10 (s, 2H), 4.74 (d, *J* = 1.1 Hz, 2H), 4.51 (t, *J* = 8.7 Hz, 2H), 3.87–3.70 (m, 4H), 3.16–2.99 (m, 2H), 2.88 (q, *J* = 7.2 Hz, 2H), 2.44 (m, 1H), 1.79 (m, 1H), 1.55 (m, 2H), 1.41–1.34 (m, 1H), 1.28–1.20 (m, 3H), 1.11 (dd, *J* = 7.6, 6.9 Hz, 3H), 0.88 (td, *J* = 12.9, 5.0 Hz, 2H). ^13^C NMR (101 MHz, CDCl_3_): δ 176.9, 159.5, 154.4, 147.7, 140.6, 139.0, 129.2, 128.1, 126.1, 119.7, 115.9, 115.9, 110.9, 106.7, 103.3, 71.0, 68.3, 59.7, 57.4, 47.7, 31.8, 29.7, 28.5, 21.4, 19.9, 14.8, 14.1, 13.3, 9.3. ESI-MS: *m*/*z* 544 [M + Na]^+^, 560 [M + K]^+^. Anal calcd for C_30_H_33_NO_7_: C 69.35, H 6.40, N 2.70, O 21.55; found C 69.31, H 6.37, N 2.73.

**(2-Ethoxy-7-(prop-2-yn-1-yloxy)benzo[d][1,3]dioxol-5-yl)methyl tosylcarbamate 18.** Compound **18** was prepared according to the general procedure. Time reaction: 16 h. The compound was purified through chromatography on silica gel with MPLC 0–60% gradient of EtOAc in petroleum ether (13 mg, 21% yield). ^1^H NMR (400 MHz, CDCl_3_) δ 7.85 (d, *J* = 8.3 Hz, 2H), 7.28 (d, *J* = 8.1 Hz, 2H), 6.85 (s, 1H), 6.55 (s, 1H), 6.45 (s, 1H), 4.94 (s, 2H), 4.74 (s, 2H), 3.71 (q, *J* = 7.1 Hz, 2H), 2.49 (t, *J* = 2.3 Hz, 1H), 2.41 (s, 3H), 1.23 (q, *J* = 9.0, 5.1 Hz, 3H). ^13^C NMR (101 MHz, CDCl_3_): δ 150.1, 147.6, 145.2, 140.5, 135.3, 129.7, 129.6, 129.2, 128.5, 128.4, 126.5, 119.7, 110.8, 103.3, 68.5, 59.6, 57.4, 29.7, 21.7, 14.8. ESI-MS: *m*/*z* 448 [M + H]^+^, 479 [M + Na]^+^. HRMS calcd for C_21_H_22_NO_8_S: 448.1066; found 448.1065.

**General procedure for release experiments** [31]. A 10 mM stock-solution of compounds **8a**, **8b**, **10**, **15**, **17**, and **18** was prepared in DMSO and diluted in the corresponding buffers to obtain a 1 mM solution. In the case of compound **16**, the 10 mM stock-solution was diluted in the corresponding buffer and additionally diluted with DMSO to obtain a final 0.67 mM solution in buffer/H_2_O/DMSO. For analysis at pH 7.4 and 6, phosphate buffers of 0.1 M were used (KH_2_PO_4_/K_2_HPO_4_). For pH 5.5 and 4.5, acetate buffer of 0.1 M was used (CH_3_COONa/CH_3_COOH). All compounds were mixed with the buffers at 25 °C and further incubated at 37 °C. HPLC analyses were carried out following the method reported in Appendix A every 1–3 h up to 6 h and every 6 h up to 48 h.

**Stability in plasma experiments**. Pooled human plasma, purchased from Merck (0.9 mL, 55.7 µg protein/mL); hepes buffer (1.0 mL, 25 mM, NaCl 140 mM, pH 7.4); and tested compound dissolved in DMSO (100 µL, 2.0 mM) were mixed in a test tube incubated at 37 °C under continuous mechanical agitation. At set time points (0.0, 0.25, 0.50, 1.0, 3.0, 5.0, 8.0, and 24.0 h), samples of 100 µL were taken, mixed with 400 µL of cold acetonitrile, and centrifuged at 5000 rpm for 15 min. The supernatant was collected and analysed by UV/LC-MS to monitor the amount of unmodified compound. LC analyses of plasma stability tests were performed using Agilent 1100 LC/MSD VL system (G1946C) (Agilent Technologies, Palo Alto, CA, USA) constituting a vacuum solvent degassing unit, a binary high-pressure gradient pump, an 1100 series UV detector, and an 1100 MSD model VL benchtop mass spectrometer. A MSD single-quadrupole instrument was equipped with the orthogonal spray API-ES (Agilent Technologies, Palo Alto, CA, USA). The pressure of the nebulizing gas and the flow of the drying gas (nitrogen used for both) were set at 40 psi and 9 L/min, respectively. The capillary voltage, fragmentation voltage, and vaporization temperature were 3000 V, 10 V, and 350 °C, respectively. MSD was used in the positive and negative ion mode. Spectra were acquired over the scan range *m*/*z* 100–2000 using a step size of 0.1. Chromatographic analyses were performed using a Phenomenex Kinetex EVO C18-100Å (150 × 4.6 mm, 5 μm particle size) at room temperature, at a flow rate of 0.6 mL/min, injection volume of 10 μL, operating with a gradient elution of A: water (H_2_O) and B: acetonitrile (ACN). Both solvents were acidified with 0.1% *v*/*v* of formic acid. UV detection was monitored at 254 nm. The analysis started with 0% of B, then B was increased to 80% (from t = 0 to t = 20 min), then kept at 80% (from t = 20 to t = 25 min), and finally returned to 0% of eluent B in 5.0 min.

## Data Availability

The data presented in this study are available upon request from the corresponding author.

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
