# Peer review of "A Self-Immolative Linker for the pH-Responsive Release of Amides"

_molecules, 2023, doi:10.3390/molecules28062445_

Round 1

Reviewer 1 Report

The authors have demonstrated the use of acid-responsive benzyl alcohols for masking drugs containing an amide bond by forming N-acyl carbamates. This platform has orthogonal reactivity and stability in plasma, making it a potential option for the targeted delivery of antibody-drug conjugates. But the authors say that more work needs to be done before this linker can be used with other non-standard chemotherapeutic drugs that are released by enzymes.

Comments to authors,

1. In this introduction, the authors make no mention of any specific flaws in the introduction.

2. I believe that the results could be improved by trying different approaches to increase the yield of N-acyl carbamates 8a and 8b and the enzalutamide prodrug 10. This could include modifying reaction conditions, such as using different solvents, reaction temperatures, and bases, as well as exploring alternative methods for acylation, such as using different activating agents. Is this something that has been investigated or considered, and is it applicable?

3. Include a comparison with how drugs are currently delivered and point out the pros and cons of acid-responsive benzyl alcohols.

4. Discuss the limitations of the study and potential future directions for improvement.

5. Data from in vitro or in vivo experiments, as well as other experimental methods, should be presented to prove the stability of plasma.

6. Talk about what this study means for how medicine is given and how it could be used to treat people.

7. Please consider the potential risks and ethical implications of using this drug delivery method.

8. To improve the yield, scaling up the reaction can help to increase the overall yield and productivity of the process. This can be achieved by using larger reactors and/or developing batch processes.

9. The introduction mentions that amides are important functional groups in drugs, but there are very few suitable bioreversible prodrugs or self-immolative linkers for their targeted release; could the author elaborate on this point, please?

10. The results could be improved by optimizing the synthesis conditions, such as pH and temperature, during the formation of the activated carbonate from benzyl alcohol and the acylation reaction. Furthermore, alternative linkers and activation methods could be explored to enhance the stability of N-acyl carbamates in physiological fluids, which might improve the desired potentials.

11.  It would be beneficial to include a more detailed characterization of the conjugates, such as their physical and chemical properties, in order to fully understand the behavior of the compounds in vivo.

12. To learn more about how well the conjugates might work in the real world, the release rates of the drugs could be studied in different physiological environments, such as in vivo animal models.

Reviewer 2 Report

1. What is the method of in vitro release?

2. The reference to in vitro study must be cited.

3. Why the medium of in vitro release is selected? Can you describe it?

4.  The kinetic release must be calculated and described.

5. The validation method of HPLC must be presented.

6. Why is amide (15-18) selected as a model drug?

7. Compond 15-18 may be used as a drug, right? Why?

8. The original structure of compounds 7, 9, 13, and 14 must be shown.

9. Why compounds 7, 9, 13, and 14 are chosen?

10. Where is the discussion of Table 2?

11. The discussion of all results is less, it may be added.

12 The conclusion is not clear.

Round 2

Reviewer 2 Report

It can be accepted